# Multilocus Genotyping and Intergenic Spacer Single Nucleotide Polymorphisms of *Amylostereum areolatum* (Russulales: Amylostereacea) Symbionts of Native and Non-Native *Sirex* Species

**DOI:** 10.3390/jof7121065

**Published:** 2021-12-11

**Authors:** Ming Wang, Ningning Fu, Chenglong Gao, Lixia Wang, Lili Ren, Youqing Luo

**Affiliations:** 1Beijing Key Laboratory for Forest Pest Control, Beijing Forestry University, Beijing 100083, China; 13020028768@163.com (M.W.); funingning2012@sina.com (N.F.); gaocl890907@163.com (C.G.); wlxynl@163.com (L.W.); 2Sino-French Joint Laboratory for Invasive Forest Pests in Eurasia, INRAE-Beijing Forestry University, Beijing 100083, China

**Keywords:** *Sirex-Amylostereum*, phylogeny, multilocus genotypes, co-occurrence, genetic diversity, population structure

## Abstract

*Sirex noctilio* along with its mutualistic fungal symbiont, *Amylostereum areolatum* (a white rot fungus), is an invasive pest that causes excessive damage to *Pinus* plantations in Northeast China. In 2015, *S. noctilio* were found to attack *Pinus sylvestris* var. *mongolica*, and often share larval habitat with the native woodwasp, *S. nitobei*. The objective of this study was to determine the possible origin(s) of the introduced pest complex in China and analyse the genetic diversity between *A. areolatum* isolated from invasive *S. noctilio*, native *S. nitobei* and other woodwasps collected from Europe (native range) and other countries. Phylogenetic analyses were performed using the intergenic spacer (IGS) dataset and the combined 4-locus dataset (the internal transcribed spacer region (ITS), translation elongation factor alpha 1 (*tef1*), DNA-directed ribosomal polymerase II (*RPB2*), and mitochondrial small subunit (mtSSU)) of three *Amylostereum* taxa. The multilocus genotyping of nuclear ribosomal regions and protein coding genes revealed at least three distinct multilocus genotypes (MLGs) of the fungus associated with invasive *S. noctilio* populations in Northeast China, which may have come from North America or Europe. The IGS region of *A. areolatum* carried by *S. noctilio* from China was designated type B1D2. Our results showed a lack of fidelity (the paradigm of obligate fidelity to a single fungus per wasp species) between woodwasp hosts and *A. areolatum*. We found that the native *S. nitobei* predominantly carried *A. areolatum* IGS-D2, but a low percentage of females instead carried *A. areolatum* IGS-B1D2 (MLG A13), which was presumably due to horizontal transmission from *S. noctilio*, during the sequential use of the same wood for larval development. The precise identification of the *A. areolatum* genotypes provides valuable insight into co-evolution between Siricidae and their symbionts, as well as understanding of the geographical origin and history of both *Sirex* species and their associated fungi.

## 1. Introduction

The genus *Amylostereum* comprises four species, namely the type species *A. chailletii* (Pers.:Fr.) Boid. (=*Stereum chailletii*), *A. areolatum* (Fr.) Boid. (=*S. areolatum*), *A. laevigatum* (Fr.) Boid (=*Peniophora laevigata*), and *A. ferreum* (Berk. and Curt.) Boid. and Lanq. (=*S. ferreum*). Of these species, the wood-rotting basidiomycetes *A. areolatum* and *A. chailletii* are well known for their mutualistic association with siricid woodwasps [1,2,3,4,5]. During oviposition, woodwasps inoculate a symbiotic fungus, *Amylostereum*, in the host tree, which exhibits the ability to cause serious damage and mortality of various conifer species, and is also the primary food source for developing wasp larvae [6,7,8,9,10,11,12]. However, *Sirex noctilio* F. and its symbiotic fungus, *A. areolatum*, are the only members of Siricidae that regularly invade living trees [6,13]. In Northeast China, *S. noctilio* has successfully colonized *Pinus sylvestris* var. *mongolica* as the main host in the field [14].

The *S**.noctilio*-*A. areolatum* complex is native to Europe and North Africa, where *S. noctilio* is considered to be a secondary pest of little economic importance [10,13,15]. In these regions, trees are rarely killed, and they may support a colony of woodwasps for more than one season [13,16]. By contrast, in the southern hemisphere and in Northeast China, the species has been causing extensive damage to pine plantations [11,12,13,17,18]. Notably, existing *S. noctilio* Integrated Pest Management (IPM) programs involve the use of biological control agents in combination with preventative silvicultural practices, which have exhibited obvious control effects in countries with a high degree of pest invasion [19,20].

*A. areolatum,* a saprophyte remaining in the tree after emergence of the wasp, can fruit and theoretically spread by means of basidiospores. The close association of *A. areolatum* with its woodwasp symbiont may be the reason that it rarely reproduces sexually in some areas of its native range, and is not yet known to fruit in its introduced region [21]. Several studies based on vegetative (or somatic) incompatibility have revealed that the isolates from South Africa and South America belong to the same vegetative compatibility group (VCG), and isolates from Lithuania, Sweden, Denmark and Great Britain share the same VCG [22,23,24]. The clonal lineages (groups of isolates with identical DNA fingerprinting profiles) of *A. areolatum*, primarily spreading vegetatively over wide geographic areas, have indicated the importance of insect vectors in the spread of fungi and the formation of dispersive clones or VCGs [22,24,25,26].

The introduction of *S. noctilio* into Northeast China and the confinement of this species to a rather small area in the country provides us with an opportunity to investigate the population of its fungal symbiont in detail. Despite much interest in these fungus-woodwasp symbioses, investigations on the population structure and phylogenetic relationships of the fungal symbiont of *Sirex* have been scarce in China. Wang et al. (2018) have revealed the significant, phylogenetic congruence between *Diaphorina citri* (Hemiptera: Liviidae) and *Candidatus* Carsonella ruddii, suggesting that the P-endosymbiont may facilitate investigations into the genealogy and migration history of the host [27]. Tracing the sources of the *S. noctilio*-*A. areolatum* complex could help determine the routes of its spread, and prevent future introductions. To study the introduced *S. noctilio* and its associated *A. areolatum* genotypes in China, we determined the genetic diversity of the *Amylostereum* spp. associated with *S. noctilio* and other woodwasps from Europe, and compared the observed genotypes with those of *S. noctilio*, collected from Northeast China.

The present study aims to investigate the phylogenetic relationship between *Sirex* and its fungal symbiont in different collection locations. In Northeast China, *S. nitobei* is the only native woodwasp exhibiting partly overlapping geographic distributions and host ranges with the introduced *S. noctilio* [14,28], which allows the possibility of host trees with mixed *Sirex* infestations. In regions where both woodwasps occur, native woodwasps emerge in the fall and attack *P. sylvestris* var. *mongolica*, whereas *S. noctilio* emerge earlier [14]. In some cases, the two species will overlap in the temporal niche, or in the development within host trees [14]. *S. nitobei* have been reported to carry *A. areolatum* or *A. chailletii* [29]. Previous studies have shown that the utility of IGS regions and ITS-*tef1*-*RPB2*-mtSSU genes for genotyping *A. areolatum* associated with *Sirex* woodwasps [30]. To investigate these sequences of the *Sirex*-*Amylostereum* associations after the establishment of *S. noctilio*, we compared the symbiotic fungi of the invasive woodwasp *S. noctilio* with the native *S. nitobei* in sympatric (in Jinbaotun (JBT) and Yushu (YS)) and independent (in Dumeng (DM), Hegang (HG) and Linyi (LY)) distributions in China. Furthermore, these sequences allow comparison of our samples with those collected from pine trees in other continents attacked by *S. noctilio*. Here, we investigated the geographical origin of *A. areolatum* clonal lineages in the areas of Northeast China recently colonized by the *S. noctilio*-*A. areolatum* complex. We believe that such studies would help determine the extent of the level of specificity between the symbiont and the host.

## 2. Materials and Methods

### 2.1. Sample Collection and Symbiont Isolation

*Amylostereum* samples were collected from *Sirex* females between 2017 and 2020 from five sites, namely, Jinbaotun (JBT), Yushu (YS), Linyi (LY), Dumeng (DM), and Hegang (HG) (Figure 1a). The sampling sites also included the sites of co-occurrence (JBT and YS) of *S. noctilio* and *S. nitobei* in Northeast China. Data for 71 *Sirex* females (female wasps carry the fungus in internal mycangia) were sources for the *Amylostereum* species and genotypes were used in the present study.

Living *Sirex* samples were collected using two methods: (1) mature *P. sylvestris* var. *mongolica*, *P. tabuliformis,* and *P. thunbergii* with signs of infestation (e.g., resin beads in Figure 1b) were cut down in spring or early summer; portions of the trees were placed in individual mesh-bags under ambient conditions. Logs were cut into 70-cm long bolts, and the ends were waxed to prevent contamination by other wood-rot fungi, and to conserve moisture. The *S. noctilio* and *S. nitobei* that emerged from the bolts were collected daily. For method (2), samples were collected weekly from lure-based black panel traps.

*S. noctilio* causes damage to *P. sylvestris* var. *mongolica* and sometimes co-infests trees with *S. nitobei*, although emergence of the relatively short-lived adult females of these two species is often separated by at least a month in Northeast China [14]. *Sirex* species were identified using the morphological method (the color of male abdomen and female legs) described by Schiff et al. [31]. Fungal symbiont samples could be collected only from adult female *Sirex*, and mycangia were removed using tweezers; the spore masses were transferred aseptically to Petri dishes with potato dextrose agar (PDA), following the method reported by Thomsen [23]. To establish fungal cultures, the contents of mycangium were transferred to a Petri dish containing PDA supplemented with antibiotics (300 mg/L streptomycin sulfate), whereas subsequent cultures were performed on PDA without antibiotics. Cultures were incubated in the dark at 25 °C ± 1 °C for two weeks.

### 2.2. Colony Polymerase Chain Reaction (PCR) and Sequencing

Fifty microliter of lysis buffer (the Lysis Buffer for Microorganism to Direct PCR, Takara) was added to each 1.5-mL microtube. Single colonies were picked using sterile pipette tips, stirred in a microtube for about five seconds and then removed. Pipette tips were not kept in the microtube for too long to avoid effects on the lysate volume and PCR amplification. After thermal denaturation at 80 °C for 15 min (ThermoUnit, CHB-100, Shanghai, China), the supernatant was centrifuged (Thermo Fresco21, 75002425, Shanghai, China) at a low speed (3000 r/3 min), and 1–5 μL of the supernatant was taken as the template for PCR. The PCR reaction mix (T100 Thermal Cycler, Beijing, China) (25 µL) is presented in Table 1, and primer sequences are presented in Table 2. Extracted DNA was stored at 4 °C until use.

### 2.3. Multilocus Genotyping

Electrophoresis was performed to examine the amplified products (Figure 2). The PCR amplicons were sent to Beijing Ruibo Biotech Co., Ltd. (Beijing, China) and sequenced using the ABI Prism™ 3730 × l automated DNA sequencer (Applied Biosystems USA, Foster City, California). Although the internal transcribed spacer region (ITS), translation elongation factor alpha 1 (*tef1*), DNA-directed ribosomal polymerase II (*RPB2*), and mitochondrial small subunit (mtSSU) data were obtained for all species, the four loci could not always be amplified for the same isolates. DNA sequences of the amplicons generated by forward and reverse primers were used to obtain consensus sequences by using SeqMan version 7.1.0 in DNAStar Lasergene Core Suite software (DNAStar Inc., Madison, WI, USA). Sequences were aligned using ClustalW [36] with default parameters and manually adjusted using a BioEdit Sequence Alignment Editor [37]. BLAST searches were performed in GenBank to identify related sequences. The degree of mutational saturation was evaluated using the substitution saturation index *I_ss_* in DAMBE version 6 [38,39,40]. The estimates of *I_ss_* were lower than the critical value *I_ss.c_* for all datasets (*p* < 0.05). All unique sequence data were submitted to GenBank (ITS: OL307781, OL307782; *tef1*: OL410536, OL410537; *RPB2*: OL410538-OL410540; mtSSU: OL323050, OL323051).

### 2.4. Fragment Analysis of Intergenic Spacer (IGS) Types

The nuc-IGS-rDNA region between the nuclear large subunit (LSU) and the 5S gene of ribosomal RNA (rRNA) operon was amplified using PCR primers, specific for the basidiomycetes P-1 [41] and 5S-2B [42]. All the isolates identified (by looking for oidia in culture [43,44]) as *A. areolatum* were verified through direct sequencing with Applied Biosystems 3730×l DNA Analyzer (Foster City, CA, USA) at Beijing Ruibo Biotech Co., Ltd., and most isolates identified as *A. chailletii* were also verified through sequencing. The nuc–IGS–rDNA region may be present in *A. areolatum* as multiple copies. The primer IGS-intF (5′- GTTTCTTAGGGCTGTTCCAGACTTGTG-3′) included a 7-bp “pigtail” (GTTTCTT) in the 5′ end. The primer 5S-2B was labeled with a FAM fluorescent marker [45]. PCR for fragment analysis was run under the following conditions: one cycle at 94 °C for 4 min; 35 cycles of 94 °C for 50 s, 55 °C for 45 s, and 72 °C for 45 s; a final extension at 72 °C for 10 min. The PCR temperature was maintained at 4 °C until gel visualization. Samples were mixed with formamide and LIZ500 size standard and then electrophoresed with the Applied Biosystems 3730×l DNA Analyzer (Foster City, CA, USA) at Beijing Ruibo Biotech Co., Ltd.; fragment sizes were determined using GeneMarke version 2.2.0 (SoftGenetics LLC) (Appendix A).

In case of heterozygosity (at least one heterozygous site) or length heterogeneity, cloning was performed. To interpret the double product amplified for some isolates of *A. areolatum* and to obtain sequence for the whole amplified fragment, secondary PCR products were cloned for sequencing. PCR fragments were purified using the pCloneEZ-NRS-Omni AMP/HC Cloning Kit. Then, it was ligated to the TOPO-cloning vector and inserted into chemically competent cells of *Escherichia coli*. Positive colonies containing the insert were screened through PCR by using the TOPO-F and TOPO-R primers. Cloned products were precipitated and purified using the aforementioned method and sequenced using the primers TOPO-F (5′-GAGCCAGTGAGTTGATTGTG-3′) and TOPO-R (5′-CAGGAAACAGCTATGACC-3′).

### 2.5. Data Analysis

Heterozygous positions were coded using degenerate bases, according to the IUPAC-IUB nomenclature. For the phylogenetic analysis, the five newly generated sequences were combined with 17 publicly available sequences (Table 3). *A. chailletii* and *A. laevigatum* were selected as outgroups. Bayesian inference (BI) in combination with MrBayes helped in partitioning the combined datasets and concomitantly applying an independent model of evolution to each partition, because unlinked genes often have different evolutionary constraints. The combined dataset was divided into four unlinked partitions (ITS, *tef1*, *RBB2*, and mtSSU). BI analysis was performed under the Kimura 2-parameter model (Kimura, 1981) plus invariant sites (K2P + I) [46]: ITS + *RPB2*; Kimura-2-parameter (K2P): *tef1*; and the likelihood model Felsenstein 1981 plus fixed (empirical) and invariant sites (Felsenstein 1981) (F81 + F + I) [47]: mtSSU nucleotide substitution models, which were determined through ModelFinder by using the Bayesian information criterion (BIC) implemented in PhyloSuite 1.21 [48]. Stationarity (chain convergence) was accessed by examining the average standard deviations of split frequencies (0.002246). Maximum likelihood (ML) phylogenetic trees were constructed using IQ-TREE [49], implemented in PhyloSuite 1.21, with 10,000 replicates of ultrafast [50] bootstrap (UFBoot) and 1000 replicates of the Shimodaira-Hasegawa-like approximate-likelihood ratio test (SH-aLRT) [51]. Nodes receiving ML bootstrap values of ≥70% and Bayesian posterior probabilities more than 0.95 were considered significantly supported. As asexual reproduction is the predominant form of reproduction in *A. areolatum*, genetic analyses were conducted with a clone-corrected dataset. The relatedness of haplotype sequences to *A. areolatum* strains and their specificity to native or non-native *Sirex* species (Appendix A) were analyzed through hierarchical clustering analysis (Ward method) in SAS-JMP pro version 16.0.0 software (SAS Institute Inc., Cary, NC, USA); each haplotype was scored as present (1) or absent (0). The haplotype of global *A. areolatum* (IGS region) distributions was mapped using POPART [52].

## 3. Results

### 3.1. Phylogenetic Relationship in Amylostereum

The final multilocus dataset utilized for ML and BI analyses consisted of 2116 characters (ITS: 574 characters; *tef1*: 345 characters; *RPB2*: 684 characters; mtSSU: 513 characters). Of all, 1935 characters were constant, 51 characters were variable and parsimony-uninformative, and 130 characters were parsimony-informative. The topology of the ML tree was mostly congruent with that of the BI tree, and the tree derived by ML was shown in Figure 3. In the *A. areolatum* clade, a subclade of multilocus genotypes (MLGs), A13, A14, and A15, associated with the non-native *S. noctilio*-*A. areolatum* from China was well supported. For IGS-D (a subclade consisting of MLGs), A4, A16, and A17 from Japan and China were the native isolates. Three strains of *A. areolatum* collected from *S. noctilio* in China had identical mtSSU and ITS loci (Table 3).

The parts of *A. areolatum* samples from the mycangia of *S. noctilio* that provided insufficient amounts of DNA for sequencing all four loci were excluded from the analysis. A total of 5 MLGs (MLGs A13–17, unique sequence) were observed in all 26 *A. areolatum* isolates from China (Table 3), which suggested the occurrence of at least five clonal lineages of *A. areolatum* in Northeast China. These MLGs did not match perfectly with those observed in other continents.

Based on the sequences of ITS, mtSSU, *RPB2*, and *tef1*, the *A. areolatum* samples were grouped into 17 MLGs (Table 3, Figure 3). Ten of these MLGs had only one sample, whereas the remaining MLGs consisted of 2–16 samples. A1, the most widespread MLG, was observed in the *S. juvencus*, *S. noctilio*, *Urocerus albicornis*, and *U. gigas* samples collected from Denmark, Hungary, or Spain. The other two widespread MLGs, A5 and A15, were also detected; A5 was observed in the *S. juvencus* and *S. noctilio* samples collected from five countries, including Europe, the United States, and Australia, whereas A15 was detected in the *S. noctilio* sample only collected from China, which is a unique and dominant haplotype not found elsewhere in the rest of the world. Comparison between DNA sequences of these MLGs, and those of the samples from pine trees invaded by *S. noctilio* in other countries, revealed that the MLGs A13, A14 and A15 (China) were highly similar to the MLG A8 reported by Castrillo et al. (2015) in the US and the MLG A7 reported in European samples. The two strains of *A. areolatum* (MLGs A16 and A17) collected from *S. nitobei* in China revealed identical *RPB2* and mtSSU loci, and alignments showed the presence of these gene haplotypes in the Japanese samples (MLG A4) (Table 3, Figure 3).

### 3.2. Haplotype Relationships Based on IGS Sequences and Fragment Analysis Data

Two different sequences (hereafter referred to as sequence B and D) for the nuc–IGS–rDNA region were observed in the *A. areolatum* isolates from China. The heterogeneity in the IGS region sequences of the *A. areolatum* isolates was not observed in those of the *A. chailletii* isolates. The IGS region of *A. areolatum* sequences was obtained from 18 strains, with each strain carrying a single haplotype. According to a panel of rDNA intergenic spacer–single nucleotide polymorphisms (haplotype-specific markers) focused on the nucleotide position in the range from 206 to 276, those were designated as type D2 (Appendix A) [53]. Fragment analysis of these strains also revealed only one peak of approximately 470 bp. The remaining 28 of the *A. areolatum* strains had heterogenic sequences, and their combination haplotypes were determined to be type B1D2 (two peaks of approximately 491 bp and 470 bp). The IGS type was determined based on published sequences in the GenBank database and through fragment analysis of strains, with either one or a combination of two haplotypes in other countries (isolates from Nielsen et al., 2009, Wooding et al., 2013, Olatinwo et al., 2013, Rabiu et al., 2013 [53,54,55,56]).

Cluster analysis of the relatedness of haplotype sequences for *A. areolatum* strains and their specificity to native or non-native *Sirex* species (Figure 4, Appendix A) showed that the haplotype D2 was associated with the native *S. nitobei* isolates (M1 and L29) in China. IGS-BE and E were found only in *A. areolatum* from *S. nigricornis* (native to North America). Other haplotypes were found in *A. areolatum*, associated with *S. noctilio*. The haplotype D2 was consistently associated with the haplotype B1 found in *A. areolatum* from non-native *S. noctilio* (except for the isolate GR94-11_IGS-B1D1 from New York, United States). The loci of ITS and mtSSU consistent with IGS haplotype were observed in all *A. areolatum* isolates (D3, D18, D10, M1 and L29) from China (Table 3, Figure 4).

### 3.3. Co-Infestation of S. noctilio and S. nitobei in Pinus

Among the *Sirex* species that emerged, *S. noctilio* and *S. ni**tobei* co-occurred in 41.4% of the 29 pine trees. For the 12 trees where co-occurrence within sections of trees could be determined, 74.7% (*n* = 269) of *S. nitobei* emerged from sections of the trees where *S. noctilio* emerged, and 41.2% (*n* = 466) of *S. noctilio* emerged from sections of the trees where *S. nitobei* emerged. *S. nitobei* mainly carried either IGS-D2 *A. areolatum* or *A. chailletii*. All *S. noctilio* females emerging from *P. sylvestris* var. *mongolica* tree carried *A. areolatum* IGS-B1D2 in different sites (DM, HG, JBT, YS) (Figure 5). Two female *S. nitobei* from the same sections of co-infested trees carried *A. areolatum* IGS-B1D2 in JBT (Table 4).

## 4. Discussion

*S. noctilio* was first found in the samples collected from Northeast China in 2013 [57]. Based on COI sequence, Sun et al. (2020) have reported the genetic diversity and structure of *S. noctilio* populations [58]. In some regions, *S. noctilio* and *S. nitobei* can infest the same trees [14]. To better understand the phylogenetic relationships within *A. areolatum* in China, we analyzed sequences from 27 strains (59%) of *A. areolatum* isolates from sympatric distributions, plus 19 strains (41%) of isolates from independent distributions. Isolates from the native and exotic countries were included to determine the patterns of geographical distribution and the origin of the introduced pest species in China.

### 4.1. Multilocus Genotyping and IGS Heterogeneity of A. areolatum

Multilocus sequencing of the *A. areolatum* isolates revealed three distinct *S. noctilio*-associated *A. areolatum* MLGs in Northeast China (Table 3), which have not been detected in *S. noctilio*-native countries. The MLGs A13, A14, and A15 exhibited sequence similarities with A8 (AH1-17) and A7 (B1385) reported by Castrillo et al. (2015), indicating that multiple *S. noctilio* species might have been introduced to China from Europe or North America (Figure 3, Table 3) [59]. These species originated probably from unrepresented source populations. Our results are consistent with the patterns of multiple invasions and the spread of *S. noctilio* across China [58]. Additional samples and heterogeneous genetic markers for assessing the population genetics of *A. areolatum* are required to determine whether these introductions occurred either through a direct route from Europe, or indirectly from North America, or a combination of both routes.

In addition to the observation of fragment size differences among the *A. areolatum* IGS rRNA sequences, we also paid attention to single nucleotide polymorphisms (SNPs). Two different-sized PCR products indicated the presence of a heterogenic sequence in the IGS of the nuclear rDNA locus in the *A. areolatum* isolates. This finding was confirmed by cloning and sequencing these fragments. We found only one genotype of *A. areolatum* in Northeast China was associated with *S. noctilio*, that is, the heterogeneous strain (BD), in contrast with the invasion of the southern hemisphere (AB). The strain carried the B sequence, previously reported in Europe and the southern hemisphere [42], along with the D sequence. IGS-BD is widely distributed across five continents (Figure 6). The homogeneous strain (D) of *A. areolatum* associated with *S. nitobei*, which is found only in the northern hemisphere, was equivalent to an isolate obtained from *S. nitobei* in Japan (B1395) [59].

### 4.2. Sirex noctilio and S. nitobei-A. areolatum Association

In this study, *S. nitobei* was found to carry *A. areolatum*-D, which were all collected from outside the known geographic repartition of *S. noctilio* (in Linyi), while a very small percentage of individuals carried *A. areolatum*-BD in sympatric distribution (*S. noctilio* and *S. nitobei*) and shared the same MLG A13 with *S. noctilio*. Moreover, *S. noctilio* and *S. nitobei* were sometimes observed as co-occurring on *P. sylvestris* var. *mongolica*, thereby providing the potential for horizontal transmission (*S. noctilio* share larval habitat with the native woodwasp when infesting the same trees) of fungal symbionts from sympatric *S. noctilio*. In addition, studies have shown the evolution of associations between these woodwasps and fungal symbionts allows plasticity [60,61,62]. Ann et al. (2018) found the spillover of *A. areolatum* from invasive *S. noctilio* to native *Urocerus* spp. [63], and the horizontal transmission of fungal strains from *S. noctilio* to *S.*
*nigricornis* [45] in North America. In Spain, the newly introduced *U. albicornis* (well known to be associated with *A. chailletii*) was found to carry a European strain, *A. areolatum* [63]. Introduced fungi would definitely have vectored to new trees during oviposition by new woodwasp hosts; thus, horizontal transmission in associations would assist in the establishment and spread of the newly introduced wood-decay fungus.

According to Slippers et al. (2002) [42], the divergence of *A. areolatum* can be attributed to the obligate relationship of *A. areolatum* with its insect vectors, and to the predominance of asexual reproduction compared with other *Amylostereum* species. This study shows that the *A. areolatum* (MLG A15) isolates from China, collected over four years (from 2017 to 2020), represent a lineage of MLG A15 linked with the IGS-BD. There is no sequence variation in the nuclear IGS rRNA region. The results indicated that vegetative reproduction in its symbiosis with *S. noctilio* was the predominant or only form of reproduction in *A. areolatum* in northern China. Basidiocarps of *A. areolatum* have not been reported in these areas. In the ecology of *Amylostereum* spp., sexual reproduction could allow genetic recombination, leading to polymorphisms. However, asexual reproduction of *A. areolatum* and vertical transmission across generations by the woodwasp vector would result in the extensive spread of cloned fungus [64]. The sample size of the present study was insufficient for fully assessing the population genetics of the *A. areolatum* associated with *S. noctilio* in China. The close similarity of A14 and A15 from *S. noctilio* with A13 from *S. noctilio* and *S. nitobei*, and its associated IGS type in Northeast China, warrants further examination of *A. areolatum* genotypes associated with more native siricid species and additional European and North American samples.

### 4.3. Focus for Further Research

Symbiont fidelity is the main mechanism in the evolution and stability of mutualisms. Heterogenic sequences in the nuc-IGS-rDNA region of *A. areolatum* isolates make it possible to compare and characterize populations of these fungi that are associated with different wasp species. The occurrence and combination of these sequences provides insight into both the geographical distribution and evolutionary relationships of populations of fungus. Effective IPM strategies rely on determining the species and genetic variants present in a given population or region, especially where both native and invasive species are co-infesting. This study can fill in this knowledge gap by characterizing the population structure of *A. areolatum* in Northeast China. Our understanding of the *A. areolatum* population structure is limited, which further impedes the development and application of effective IPM approaches. Identification studies utilizing DNA sequence data on a large number of isolates from native areas, where diverse wasp species/populations co-exist, would be valuable. Such studies would make it possible to determine the extent of the levels of specificity between the symbiont and host.

## Figures and Tables

**Figure 1 jof-07-01065-f001:**
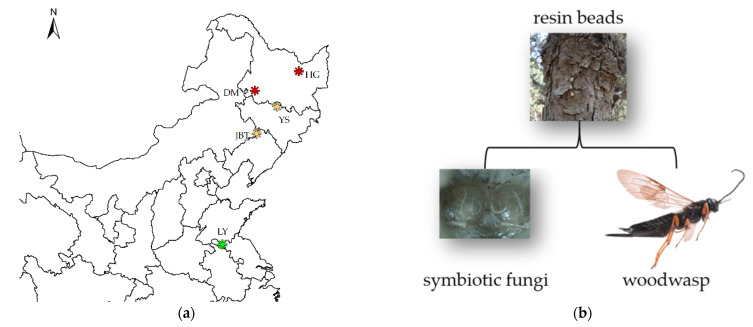
(**a**) Map of China highlighting the sampling areas (Jinbaotun (JBT), Yushu (YS), Linyi (LY), Dumeng (DM), and Hegang (HG)) in the current study; different colors represent different *Sirex* species present in China; red represents *S. noctilio*, green represents *S. nitobei*, and yellow represents co-occurring *S. noctilio* and *S. nitobei*. (**b**) Signs of infestation (resin beads) of woodwasp in the host tree and symbiotic fungi.

**Figure 2 jof-07-01065-f002:**
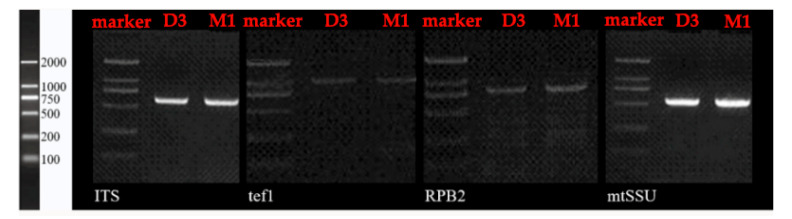
Visualization of the amplification of the four genes (the internal transcribed spacer region (ITS), translation elongation factor alpha 1 (*tef1*), DNA-directed ribosomal polymerase II (*RPB2*), and mitochondrial small subunit (mtSSU)) of *Amylostereum* isolates from *Sirex*, on a 1.5% agarose gel stained with ethidium bromide.

**Figure 3 jof-07-01065-f003:**
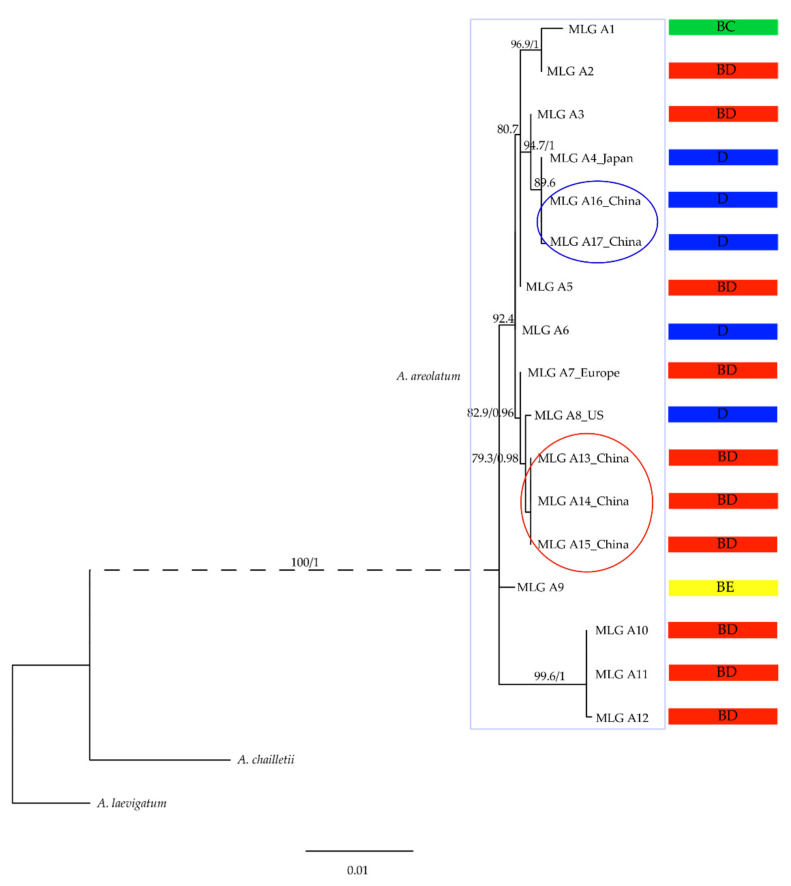
Phylogenetic relationships of *Amylostereum* spp. Majority-rule consensus tree based on the concatenated ITS, *tef1*, *RPB2*, and mtSSU genes inferred from maximum likelihood (ML) and Bayesian inference (BI) analyses. ML bootstrap values (≥70%) and Bayesian posterior probabilities (>0.95) are shown at the nodes. The IGS types are indicated on the right (BC, BD, D, BE). The Chinese strains are circled in red and blue. Clonal lineages with missing data for the *tef1* locus were excluded from the analysis.

**Figure 4 jof-07-01065-f004:**
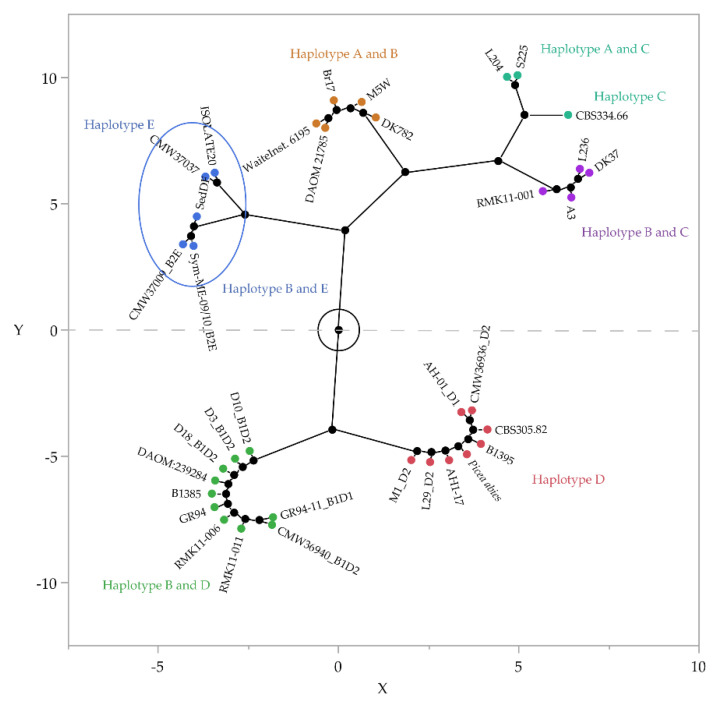
Hierarchical clustering analysis and constellation plot based on 35 isolates from the GenBank (Appendix A, unique sequence), and relatedness of haplotypes in the *Amylostereum areolatum* isolates from native and non-native *Sirex* specimens, from multiple locations in China. The plot arranges the *A. areolatum* isolates as endpoints, and each cluster join as a new point, whereas the lines represent membership in a cluster. The length of a line between cluster joins approximates the distance between the clusters that were joined, where X- and Y-axes enable comparison of the relative distance between clusters; the longer the lines, the greater the distance between the clusters. The axis scaling, orientation of points, and angles of the lines on the constellation plot are arbitrary, with no assigned unit in SAS-JMP pro version 16.0.0.

**Figure 5 jof-07-01065-f005:**
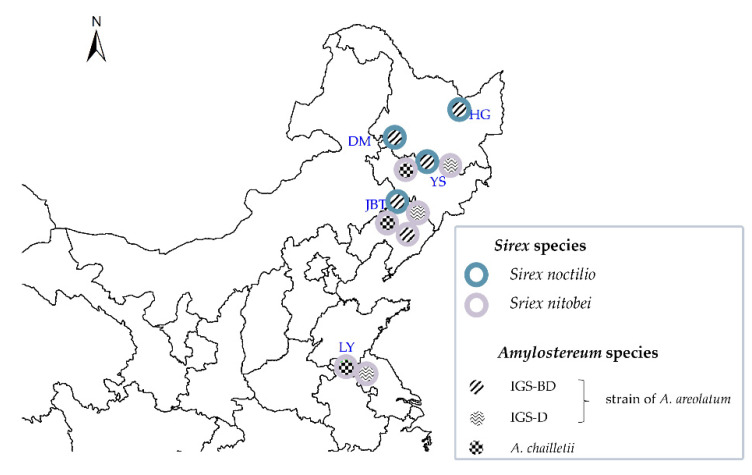
Distribution of the associations between *Sirex* species and *Amylostereum* species (strains) in China.

**Figure 6 jof-07-01065-f006:**
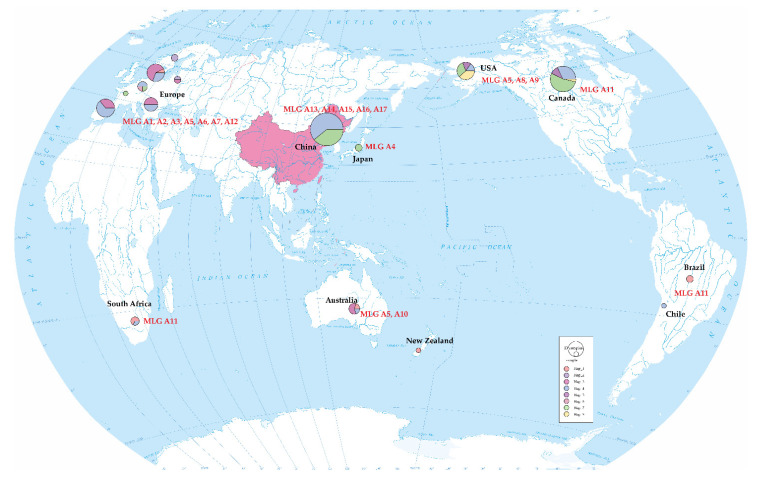
Global *A. areolatum* associated with woodwasp population haplotype distribution, based on the IGS region and MLGs.

**Table 1 jof-07-01065-t001:** Polymerase Chain Reaction (PCR) amplification system.

Composition	Dosage (μL)
The supernatant	1
Premix Taq (Ex Taq Version 2.0) 1.25 U/25 μL	12.5
Primer (F) 10 pmol/μL	1
Primer (R) 10 pmol/μL	1
Sterile distilled water	9.5

**Table 2 jof-07-01065-t002:** Primer pairs used in this study for colony PCR tests (written 5’-3’).

Target Gene	Primer Sequence (5′–3′)	Reference	PCR Amplification Conditions
ITS			95 °C 4 min; 95 °C 1 min; 55 °C 1 min; 72 °C 1 min; 72 °C 10 min; 35 cycles
ITS1-F	5′-CTTGGTCATTTAGAGGAAGTAA	Gardes and Bruns (1993) [32]	
ITS4-B	5′-CAGGAGACTTGTACACGGTCCAG		
*tef1*			95 °C 4 min; 95 °C 1 min; 55 °C 1 min; 72 °C 1 min; 72 °C 10 min; 35 cycles
*tef1*F	5′-TACAAGTGCGGTGGTATTGACA	Morehouse et al. (2003) [33]	
*tef1*R	5′-ACGGACTTGACTTCAGTGGT		
*RPB2*			95 °C 4 min; 95 °C 1 min; 50 °C 2 min; 72 °C 2 min; 72 °C 10 min; 35 cycles
*RPB2*-6F	5′-TGGGGTATGGTCTGTCCTGC	Liu et al. (1999) [34]	
f*RPB2*-7.c R	5′-TGGGGTATGGTCTGTCCTGC		
mtSSU			95 °C 4 min; 95 °C 1 min; 55 °C 1 min; 72 °C 1 min; 72 °C 10 min; 35 cycles
MS1	5′-CAGCAGTCAAGAATATTAGTCAATG	White et al. (1990) [35]	
MS2	5′-GCGGATTATCGAATTAAATAAC		

**Table 3 jof-07-01065-t003:** Multilocus genotypes based on ITS, mtSSU, *RPB2*, and *tef1* sequence data of *Amylostereum areolatum* from *Sirex* spp. and *Urocerus* spp. from various countries and sites.

MLG ^1^	Isolate Code	ITS-*tef1*-*RPB2*-mtSSU Types	Woodwasp/Tree Host	Collection Site	No. Samples	IGS
*A. areolatum*						
A1	RMK11-001	A-A-A-A	*S. juvencus, S. noctilio, U. albicornis, U. gigas*	Denmark, Hungary, and Spain	16	BC
A2	RMK11-011	B-A-A-B	*S. juvencus*	Hungary	1	BD
A3	RMK11-022	B-A-B-B	*S. noctilio*	Hungary	1	BD
A4	B1395	C-A-B-B	*S. nitobei*	Japan	2	D2
A5	GR94	B-A-C-B	*S. juvencus, S. noctilio*	Australia, Denmark, Hungary, Spain, and US	14	BD
A6	B1352	D-A-C-B	tree source (*Picea abies*)	Germany	1	D
A7	B1385	D-A-C-D	*S. juvencus*	Germany	1	BD
A8	AH1-17	B-A-C-E	*S. noctilio*	US	1	D
A9	ScyME9/10	E-B-C-C	*S. nitidus*	US	1	BE
A10	Ecogrow	F-C-D-B	Ecogrow nematode culture	Australia	1	BD
A11	DAOM:239281	D-C-D-B	*S. noctilio*	Canada, South Africa, Chile	8	BD
A12	DAOM:Francke	D-C-D-D		Germany	1	BD
A13	D18	D-D-D-E	*S. noctilio, S. nitobei*	DM, JBT (this study)	4	BD
A14	D3	D-D-E-E	*S. noctilio*	DM (this study)	1	BD
A15	D10	D-E-D-E	*S. noctilio*	DM, YS, HG, and JBT (this study)	11	BD
A16	M1	G-E-B-B	*S. nitobei*	JBT, LY, YS (this study)	9	D
A17	L29	G-D-B-B	*S. nitobei*	LY (this study)	1	D
*A. chailletii*			*U. gigas*	Australia, Denmark, Hungary, Spain, and US	1	
*A. laevigatum*		-	*U. antennatus*	Japan	1	

^1^ Each letter represents a unique sequence for each locus, and sequence data for representative strains with different types were deposited to GenBank.

**Table 4 jof-07-01065-t004:** Species and IGS strains of *Amylostereum* associated with *Si**rex noctilio* and *Sirex nitobei* in the current study.

*Sirex* Host	Samples, *n*	*A. areolatum*		*A. chailletii*
		IGS-D2	IGS-B1D2	
*Sirex noctilio*	26	-	26	-
*Sirex nitobei*	44	18	2	24

## Data Availability

Not applicable.

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
