# Peer review of "Multilocus Genotyping and Intergenic Spacer Single Nucleotide Polymorphisms of Amylostereum areolatum (Russulales: Amylostereacea) Symbionts of Native and Non-Native Sirex Species"

_jof, 2021, doi:10.3390/jof7121065_

Round 1

Reviewer 1 Report

Please find below some suggestions that would help the reader to get to your point:

Important comments:

  • End of the introduction: please explain to the reader, who is not necessarily familiar with your approach, why you have used two types of markers: a combination of ITS-tef1-RPB2-mtSSU and the IGS region.
  • Also, a clear description of the sets of strains selected for the study, and the rationale for this selection, would help understanding the conclusions of the analysis
  • L211 – Fig 3: Only one strain of chailletti was included to the analysis. To me, this does not allow to conclude that A. areolatum and A. chailletti are monophyletic
  • Fig 3 The continent is not always indicated on the figure. Indicate in the legend what is highlighted in the circle. I do not see the Sirex species indicated on the tree. I do not understand how this figure shows the “relationships of Amylostereum spp. with Sirex noctilio, S. nitobei, and other woodwasps”.

Other comments

  • L 22-23: I do not understand your point, please rephrase
  • L 24 “lack of fidelity”: please define
  • L39 “symbiotic fungal pathogen”: I guess that you mean symbiotic on wasps and pathogen on trees, please do not combine everything in 3 words.
  • L 61-64: a verb?
  • L 71: “ ruddii”: not mentoned before, what is it?
  • L74 “to study the biology”: I don’t think the genetic diversity is used to study the biology of the wasp here
  • L 79-80: 2 verbs in a sentence
  • L84 “overlap during flight”: what is that?
  • L 97-98 “females that were sources for”: please rephrase
  • L112 “the method used by…”: please indicate shortly if the method is based on morphological or molecular markers
  • L121: please indicate the composition or the brand of the lysis buffer
  • L128 : “the PCR reaction mix is presented in Table 1”
  • Table 1: please indicate the units for the enzyme, and concentration for the primers
  • Table 2: the PCR cycles can be provided in the text, only the annealing temperature should be indicated in the table.
  • L140 “data were obtained for all species”, “…not always amplified..”
  • L 141 “DNA sequences of the amplicons ..”
  • Fig2 please indicate what was loaded on the gels
  • L155 “The nuc-IGS-rDNA region…”
  • L157 “the isolates identified as areodatum…”: what are the criteria for this identification?
  • L163: “… added to the 5’ end” of …??
  • L167 “LIZ5003” brand?
  • L171 “more than one heterozygous site”: I guess you mean “at least one”?
  • L173 what is a “secondary PCR product”?
  • L175 please indicate that you have done coli transformation before screening for positive colonies
  • L211-314: 2 verbs in a same sentence
  • L220: “… were observed in all 26 areolatum isolates from China”
  • L222: “… those observed in other continents”
  • Table 3: the reader that is not familiar on the topic cannot understand why Urocerus is included in the study. What is “Ecogrow”?
  • L250-251: this is not shown, could be removed I suppose
  • L 254: what is the “IGS-SNP panel analysis”?
  • L 245 “fragment analysis”. Do you mean « amplicon length”?
  • L 283-256: what are those 18 and 28 strains? There are 48 + 26 strains on Table 3
  • L257: It would help if you remind/explain to the reader what is a “heterogenic sequence”, vs “heterozygous sequence”.
  • L273-274: what are “M1 and L29”?
  • L273-279: it is very difficult to get to your point. A proposition would be to add IGS results in Table 3.
  • L284-287: I don’t understand what you mean
  • L288 “carried IGS-B1D2 in different sites”: please rephrase
  • L296: “ … we analyzed the sequences from 48 strains (64%) … , plus 26 isolates (36%)…”
  • L312 what do you mean with “fragment size difference”
  • L328 “the known geographic repartition of”

Author Response

Thank you for your comments on our manuscript entitled " Multilocus genotyping and intergenic spacer single nucleotide polymorphisms of Amylostereum areolatum (Russulales: Amylostereacea) symbionts of native and non-native Sirex species " (ID: insects-696716). Those comments are very helpful for revising and improving our paper, as well as the important guiding significance to our research. We have studied the comments carefully and made corrections which we hope meet with approval. The main corrections are in the manuscript and the responds to the reviewers’ comments are as follows (the replies are highlighted in red).

Dear Editor,

We sincerely appreciate you for giving us the opportunity to revise our manuscript, sending us the reviewers’ suggestions and editorial requests concerning our manuscript entitled "Multilocus genotyping and intergenic spacer single nucleotide polymorphisms of Amylostereum areolatum (Russulales: Amylostereacea) symbionts of native and non-native Sirex species" (ID: jof-1492832). Those comments are all quite valuable and very helpful for revising and improving our paper. We have gone through every detail of the revision list very carefully and have revised our manuscript according to the reviewers’ suggestions and editorial requests. We hope that our modifications to the manuscript could address all the questions of reviewers and editors. Responses to reviewers’ suggestions are listed as blow. Revised portion are marked in red in the manuscript. And once again, thanks for all your dedicated work.

Best regards!

Sincerely yours,

Lili Ren, Youqing Luo, Ming Wang, Ningning Fu, Chenglong Gao, Lixia Wang.

Response to reviewers’ suggestions

Thank you for your comments on our manuscript entitled " Multilocus genotyping and intergenic spacer single nucleotide polymorphisms of Amylostereum areolatum (Russulales: Amylostereacea) symbionts of native and non-native Sirex species " (ID: insects-696716). Those comments are very helpful for revising and improving our paper, as well as the important guiding significance to our research. We have studied the comments carefully and made corrections which we hope meet with approval. The main corrections are in the manuscript and the responds to the reviewers’ comments are as follows (the replies are highlighted in red).

Reviewer #1

Thank you very much for your valuable comments and suggestions. These comments and suggestions have been of great help to our research, not only to make our manuscripts more scientific and enriching, but also to broaden our horizons. In addition, we have asked several colleagues who are native speaker to check the English. We believe that the language is now acceptable for the review process.

Point 1: End of the introduction: please explain to the reader, who is not necessarily familiar with your approach, why you have used two types of markers: a combination of ITS-tef1-RPB2-mtSSU and the IGS region.

Response 1: Thank you very much for your suggestion. We have followed a study by Bergeron et al. (2011) showing the utility of these regions and genes for genotyping A. areolatum associated with Sirex woodwasps. Further, these sequences would allow comparison of our samples with those collected from pine trees in other countries attacked by Sirex noctilio.

Point 2: Also, a clear description of the sets of strains selected for the study, and the rationale for this selection, would help understanding the conclusions of the analysis.

Response 2: Thank you very much for your suggestion. We modified the sentence to clarify the two purposes of selecting the set of strains: comparing symbiotic fungus carried by S. noctilio in different continents (possible origin of A. areolatum populations); comparing of symbiotic fungi carried by invasive species and native woodwasps (horizontal transmission between woodwasps).

Point 3: L211 – Fig 3: Only one strain of A. chailletti was included to the analysis. To me, this does not allow to conclude that A. areolatum and A. chailletti are monophyletic.

Response 3: Thank you very much for your suggestion. We have deleted this sentence.

Point 4: Fig. 3 The continent is not always indicated on the figure. Indicate in the legend what is highlighted in the circle. I do not see the Sirex species indicated on the tree. I do not understand how this figure shows the “relationships of Amylostereum spp. with Sirex noctilio, S. nitobei, and other woodwasps”.

Response 4: We have modified the legend. Details of the continents and relationships of Amylostereum spp. with Sirex noctilio, S. nitobei, and other woodwasps were given in Table 3.

Points:

- L 22-23: We have rephrased the sentence “The IGS region of A. areolatum carried by S. noctilio from China was designated type B1D2.”. Thank you very much.

- L 24 “lack of fidelity”: The paradigm of obligate fidelity to a single fungus per wasp species

- L39: We have revised it in the manuscript. Thank you very much.

- L 61-64: We revised the sentence “The clonal lineages (groups of isolates with identical DNA fingerprinting profiles) of A. areolatum, primarily spreading vegetatively, over wide geographic areas indicated the importance of insect vectors in the spread of the fungi and formation of dispersive clones or VCGs.”.

- L 71: We have revised it in the manuscript. It is another example of symbiont that may facilitate investigations into the genealogy and migration history of the host.

- L74: Thank you very much for your suggestion. We have deleted that.

- L 79-80: We have revised it in the manuscript.

- L84: The 2 species will overlap in the temporal niche.

- L 97-98: Female wasps carry the fungus in internal mycangia.

- L112: Based on morphological methods (the color of male abdomen and female legs).

- L121: We have added the brand of the lysis buffer.

- L128: It has been modified in the paper.

- Table 1: We have added the units for the enzyme and concentration for the primers.

- Table 2: We have revised it in table 2.

- L140: We have revised words in the manuscript.

- L 141: We have added the word.

- Fig2: We have added the marker and the name of strains.

- L155: We have changed "For" to "The". Thank you very much.

- L157: Distinguishing A. areolatum from A. chailletii is most easily done by looking for oidia in culture.

- L163: We have revised the sentence in the manuscript. Thank you very much.

- L167: “LIZ5003” is internal lane standards.

- L171 We have changed "more than one" to "at least one". Thank you very much.

- L173: PCR products were used as templates for secondary PCR.

- L175: We have added E. coli transformation in the manuscript.

- L211-214: We have revised the sentence in the manuscript.

- L220: We have added “from China” in the manuscript.

- L222: We have replaced “Europe” with “other continents” in the manuscript.

- Table 3: The same MLG was shared with the S. noctilio and native Urocerus spp. population. Ann et al. (2018) found the spillover of A. areolatum from invasive S. noctilio to native Urocerus spp.

“Ecogrow”: One isolate of A. areolatum that accompanied the parasitic nematode Deladenus siricidicola, purchased from Ecogrow (Bondi Beach, New SouthWales, Australia).

- L250-251: We have deleted the sentence.

- L 254: A panel of rDNA intergenic spacer–single nucleotide polymorphisms.

- L 245: Yes, it means amplicon length.

- L 283-256: Because clonal lineages with missing data for the tef1 locus were excluded from the analysis (18 and 28 strains). 44 + 26 strains were only baesd on IGS sequences.

- L257: Isolates of A. areolatum contained different combinations of heterogenic sequences (B/D). Heterogenic sequence included at least one heterozygous site/position.

- L273-274: That’s in Figure 4: M1_D2 and L29_D2.

- L273-279: We have added IGS results in Table 3.

- L284-287: In this study, we investigated the percentages of these now-sympatric native and invasive siricids utilizing the same sections of co-infested suppressed pines, providing a possibility for horizontal transmission of fungal strains from S. noctilio to S. nitobei.

- L288: We have revised the sentence in the manuscript.

- L296: We have revised the sentence in the manuscript.

- L312: It indicated length of IGS amplicons.

- L328: We have revised the sentence in the manuscript.

We tried our best to improve the manuscript and made some changes in the manuscript. These changes will not influence the content and framework of the paper. And here we did not list these changes.

We appreciate for Reviewers’ warm work earnestly, and hope that the correction will meet with approval. Once again, thank you very much for your comments and suggestions.

Reviewer 2 Report

Comments to authors:

regarding the writing format of the genetic markers, tef and rpb should be in italic due to are protein enconding genes. Conversely the ITS and IGS regions is does not code for protein so, must not be written in italics.

In table 2, specifically in the fourth columna replace “annealing temperatures” for “PCR amplification conditions”

Line 188 and 189: should be indicated the complete name of each substitution model (K2P+I, K2P and F81+F+I), and these abreviations in Brackets.

In the item 2.5 Data análisis, I recommend indicating the ML bootstrap values considered significant (in this case bs ≥ 70%) and for BI indicate the posterior probability (pp) value considered significant (normally ≥ 0.95).

Line 205: the final multilocus dataset (ITS, tef1, rpb2 and mtSSU), is composed by a total of 2101 characters instead of 2116 indicated by the authors, check it.

In the figure 3: the legend of the figure 3 indicate that the tree is made with ML and BI analyses, however the nodes only indicate the ML bootstrap values. The authors should indicate the Bayesian posterior probabilities in the nodes, using the following formate: MB Bootstrap > 70% / Bayesian posterior probabilities > 0.95.

In the line 247: replace “baesd” for “based”

Regarding to the references, review the format according to the instructions of the journal:

  • Line 415: Replace “SPRADBERY” by “Spradbery”.
  • Line 417: replace “BIONOMICS OF SIRICIDAE” by “Bionomics of Siricidae”.
  • Line 422, 424, 430: the year of publication should be in bold letter.
  • Reference 27: cite correctly the name and surname of the authors.
  • Standardize the máximum number of authors according to the journal format.

Author Response

Thank you for your comments on our manuscript entitled " Multilocus genotyping and intergenic spacer single nucleotide polymorphisms of Amylostereum areolatum (Russulales: Amylostereacea) symbionts of native and non-native Sirex species " (ID: insects-696716). Those comments are very helpful for revising and improving our paper, as well as the important guiding significance to our research. We have studied the comments carefully and made corrections which we hope meet with approval. The main corrections are in the manuscript and the responds to the reviewers’ comments are as follows (the replies are highlighted in red).

Dear Editor,

We sincerely appreciate you for giving us the opportunity to revise our manuscript, sending us the reviewers’ suggestions and editorial requests concerning our manuscript entitled "Multilocus genotyping and intergenic spacer single nucleotide polymorphisms of Amylostereum areolatum (Russulales: Amylostereacea) symbionts of native and non-native Sirex species" (ID: jof-1492832). Those comments are all quite valuable and very helpful for revising and improving our paper. We have gone through every detail of the revision list very carefully and have revised our manuscript according to the reviewers’ suggestions and editorial requests. We hope that our modifications to the manuscript could address all the questions of reviewers and editors. Responses to reviewers’ suggestions are listed as blow. Revised portion are marked in red in the manuscript. And once again, thanks for all your dedicated work.

Best regards!

Sincerely yours,

Lili Ren, Youqing Luo, Ming Wang, Ningning Fu, Chenglong Gao, Lixia Wang.

Response to reviewers’ suggestions

Thank you for your comments on our manuscript entitled " Multilocus genotyping and intergenic spacer single nucleotide polymorphisms of Amylostereum areolatum (Russulales: Amylostereacea) symbionts of native and non-native Sirex species " (ID: insects-696716). Those comments are very helpful for revising and improving our paper, as well as the important guiding significance to our research. We have studied the comments carefully and made corrections which we hope meet with approval. The main corrections are in the manuscript and the responds to the reviewers’ comments are as follows (the replies are highlighted in red).

Reviewer #2

Thank you very much for your valuable comments and suggestions. These comments and suggestions have been of great help to our research, not only to make our manuscripts more scientific and enriching, but also to broaden our horizons.

Point 1:               Regarding the writing format of the genetic markers, tef and rpb should be in italic due to are protein enconding genes. Conversely the ITS and IGS regions is does not code for protein so, must not be written in italics.

Response 1: Thank you very much for your suggestion. We have re-written the genetic markers according to reviewers’ suggestions.

Point 2: In table 2, specifically in the fourth columna replace “annealing temperatures” for “PCR amplification conditions”

Response 2: We have changed " annealing temperatures " to " PCR amplification conditions ". Thank you very much.

Point 3: Line 188 and 189: should be indicated the complete name of each substitution model (K2P+I, K2P and F81+F+I), and these abreviations in Brackets.

Response 3: We have added the complete name of substitution model in manuscript.

Point 4: In the item 2.5 Data análisis, I recommend indicating the ML bootstrap values considered significant (in this case bs ≥ 70%) and for BI indicate the posterior probability (pp) value considered significant (normally ≥ 0.95). 

Response 4: We have added values in paper. Nodes receiving ML bootstrap values of ≥70 % and Bayesian posterior probabilities more than 0.95 were considered significantly supported.

Point 5: Line 205: the final multilocus dataset (ITS, tef1, rpb2 and mtSSU), is composed by a total of 2101 characters instead of 2116 indicated by the authors, check it.

Response 5: We have checked the total number of characters in manuscript.

Point 6: In the figure 3: the legend of the figure 3 indicate that the tree is made with ML and BI analyses, however the nodes only indicate the ML bootstrap values. The authors should indicate the Bayesian posterior probabilities in the nodes, using the following formate: MB Bootstrap > 70% / Bayesian posterior probabilities > 0.95.

Response 6: We have added the Bayesian posterior probabilities in the nodes to the figure.

Point 7: In the line 247: replace “baesd” for “based”

Response 7: We have replaced “baesd” for “based”.

Point 8: Regarding to the references, review the format according to the instructions of the journal:

  • Line 415: Replace “SPRADBERY” by “Spradbery”.
  • Line 417: replace “BIONOMICS OF SIRICIDAE” by “Bionomics of Siricidae”.
  • Line 422, 424, 430: the year of publication should be in bold letter.
  • Reference 27: cite correctly the name and surname of the authors.
  • Standardize the máximum number of authors according to the journal format.

Response 8: We have corrected the references in the pater.

We tried our best to improve the manuscript and made some changes in the manuscript. These changes will not influence the content and framework of the paper. And here we did not list these changes.

We appreciate for Reviewers’ warm work earnestly, and hope that the correction will meet with approval. Once again, thank you very much for your comments and suggestions.